

# The Hydrological Archetypes of Wetlands

Abigail E. Robinson[1], Anna Scaini[1], Francisco J. Peña[1,2], Peter A. Hambäck[3], Christoph Humborg[1], Fernando Jaramillo[1]

[1]Department of Physical Geography, Stockholm University, Stockholm, 106 91, Sweden
[2]Department of Medical Epidemiology and Biostatistics, Karolinska Institutet, Stockholm, 171 77, Sweden
[3]Department of Ecology, Environment and Plant Sciences, Stockholm University, Stockholm, 106 91 Sweden

*Correspondence to*: Abigail E. Robinson (abigail.robinson@natgeo.su.se)

**Abstract**

Wetlands are valuable and diverse environments that contribute to a vast range of ecosystem services, such as
flood control, drought resilience, and carbon sequestration. The provision of these ecosystem services depends on their hydrological functioning, which refers to how water is stored and moved within a wetland environment. Since the hydrological functions of wetlands vary widely based on location, wetland type, hydrological connectivity, vegetation, and seasonality, there is no single approach to defining these functions. Consequently, accurately identifying their hydrological functions to quantify ecosystem services remains challenging. To address this issue,
we investigate the hydrological regimes of wetlands, focusing on water extent, to better understand their hydrological functions. We achieve this goal using Sentinel-1 SAR imagery and a self-supervised deep learning model (DeepAqua) to predict surface water extent for 43 Ramsar sites in Sweden between 2020-2023. The wetlands are grouped into the following archetypes based on their hydrological similarity: 'autumn drying', 'summer dry', 'spring surging', 'summer flooded', 'spring flooded' and 'slow drying'. The archetypes represent great
heterogeneity, with flashy regimes being more prominent at higher latitudes and smoother regimes found preferentially in central and southern Sweden. Additionally, many archetypes show exceptional similarity in the timing and duration of flooding and drying events, which only became apparent when grouped. We attempt to link hydrological functions to the archetypes whereby headwater wetlands like the spring-surging archetype have the potential to accentuate floods and droughts, while slow-drying wetlands, typical of floodplain wetlands, are more
likely to provide services such as flood attenuation and low flow supply. Additionally, although wetlands can be classified in myriad ways, we propose that classifying wetlands based on the hydrological regime is useful for identifying hydrological functions specific to the site and season. Lastly, we foresee that hydrological regime-based classification can be easily applied to other wetland-rich landscapes to understand the hydrological functions better and identify their respective ecosystem services.





## 1 Introduction

Wetlands are ecosystems that are seasonally or permanently covered by or saturated with water (Bullock and Acreman, 2003). After centuries of wetland loss (Fluet-Chouinard et al., 2023), wetlands are now viewed as key providers of provisioning and regulating services such as forestry, fishing, food production, flood control, drought resilience, nutrient and sediment retention and carbon sequestration (Ameli and Creed, 2019; Barbier et al., 1997; Colvin et al., 2019; Johnston, 1991; Matthew et al., 2010; Tang et al., 2020; Villa and Mitsch, 2015). Additionally,

they offer cultural and supporting services (Margaryan et al., 2022; Mitsch et al., 1991) and are crucial for achieving the sustainable development goals outlined in Agenda 2030 (Jaramillo et al., 2019).

  The degree to which wetland environments provide ecosystem services is largely controlled by their hydrological

functions (Okruszko et al., 2011) or how wetlands store and transfer water. For instance, hydrological functions such as prolonged water storage contribute to services like flood control and sustaining water supply during low flow periods (Åhlén et al., 2020; Bullock and Acreman, 2003; Gerakēs, 1992). Other functions, such as surface-ground water exchange, relate to provisioning services such as water supply, while surface wetness and soil moisture help regulate the local climate and retain nutrients (Ameli and Creed, 2017; Hansson et al., 2005; Le and

Kumar, 2014; Mitsch et al., 2015). Furthermore, surface water extent variability is strongly correlated to fluctuations of methane emissions for boreal wetlands (North of 50°N), which is important for carbon cycling (Ringeval et al., 2010). Quantifying the hydrological functions of any wetland and the ecosystem services it provides is challenging as wetlands are spatiotemporally variable and diverse (McLaughlin and Cohen, 2013). For example, a wetland type can either reduce or enhance flooding downstream depending on the environmental setting

or time of year (Bullock and Acreman 2003). One way to improve our understanding of wetland hydrological functions and related ecosystem services is by quantifying their hydrological regime. This refers to the seasonal availability of water (water, extent, or volume) within a wetland, measured through either in-situ or remote sensing technologies (Acreman and Holden, 2013; Helmschrot, 2016).

The analysis of hydrological regimes to understand hydrological functioning usually focuses on rivers and catchments (Magilligan and Nislow, 2005; Robinson and Sivapalan, 1997). However, over the last two decades, its application for wetlands has steadily increased (e.g., Cuevas et al., 2024; Stevaux et al., 2020; Na and Li, 2022; Vilardy et al., 2011). In fact, methods for studying water extent have been driven by the need to quantify ecosystem services (Park et al., 2022). For instance, by monitoring certain water level or extent thresholds throughout the

year, we can evaluate whether a wetland is in a water-storing or transmitting state, which influences its ability to



attenuate high flows downstream (Spence et al., 2011; Yanfeng and Guangxin, 2021). For example, analysis of the hydrological regimes of wetlands in Siberia has enhanced the understanding of how their early-year (winter) water availability influences their contribution to spring flooding (Zakharova et al., 2014). In Europe, Vera-Herrera et al. (2021) demonstrated that grouping wetlands based on their long-term changes can help to maximize 65 agricultural productivity, while Åhlén et al (2022) distinguished between flood buffering capacity of wetlands in upland and downstream wetlands using variations in water level.

When in-situ water level measurements from water gauges are spatiotemporally sparse, water extent changes can be used to understand the hydrological regime. Estimating hydrological regimes from water extent is achievable 70 with remote sensing technologies, such as optical or Synthetic Aperture Radar (SAR) (Graversgaard et al., 2021; Ramsar Convention, 2011; Vera-Herrera et al., 2021). For example, multi-spectral optical sensors like Sentinel-2 can estimate surface water extent at a resolution of 10 m (Brown et al., 2022). Others have exploited the ability of SAR to detect water below flooded vegetation in a range of wetland environments at similar resolutions (Canisius et al., 2019; Kovacs et al., 2013; Melack and Hess, 2011; Widhalm et al., 2015; Peña et al., 2024).


It is widely recognised that although ecosystem services are not undervalued, they are often poorly characterised and understood in the context of wetlands. Furthermore, generalising hydrological functions and services across different wetlands is not recommended due to their unique characteristics. Here, we quantify changes in water surface extent to understand the hydrological regimes of wetlands and determine their hydrological functions, 80 using the case of the extended set of wetlands under the Ramsar Convention in Sweden. Doing so would help quantify their ecosystem services (unknown to date), particularly emphasising hydrology-based services such as flood attenuation and low flow supply.

We use a remote sensing approach to categorise wetlands by their hydrological regime based on recent water extent 85 observations (from 2020 to 2023) using a pre-trained self-supervised deep learning model called DeepAqua, which has been used to detect water from Sentinel-1 imagery (Peña et al., 2024). We use the case of 43 Ramsar wetlands as they are well inventoried, present good coverage of SAR data, and are of national and international importance due to the ecosystem services they provide (Gunnarsson and Löfroth, 2014; Ramsar Convention, 2011). We also propose that by grouping hydrologically similar sites into descriptive archetypes (as suggested by Lane et al., 90 2018), more comprehensive insights can be gained about the hydrological regime (and thus functions) than by studying each wetland's hydrological regime in isolation.



## 2 Methods

### 2.1. Wetland dataset description

Sweden has 68 Ramsar wetlands in total (Ramsar Convention, 1971). Here, we excluded coastal sites because
coastal wetlands are hydrologically different from inland wetlands and, thus, should be studied separately. Sites
with a total area exceeding 180,000 ha were also excluded due to computational and memory limitations when
computing water extent changes with deep learning. Lastly, sites with poor SAR data availability were omitted
from the analysis. This left 43 Ramsar sites suitable for hydrological regime analysis, and each site was delimited
based on the boundaries of the Ramsar Convention (Ramsar Convention - Sweden, 2023) (Fig. 1).


The sites are distributed throughout all regions in Sweden, albeit with a higher concentration of sites towards the
south. Site areas range between 200 ha and 28 900 ha and encompass various wetland types, including marshes,
fens, bogs, mires, palsa mires, lakes, streams, wetland forests, lakes, peatlands, and shrub wetlands. For these
wetlands, during the observation period (2020-2023), the average temperature and precipitation were 5.76°C and
706.5 mm, which was 0.68°C warmer and 25.6 mm wetter on average compared to the 1990-2020 climate normal
(Johansson, 2002). Additionally, the mean number of snow days in Sweden between 2020-2023 was 108.0, which
is 12.3 days less compared to the last climate normal (Climate indicator - Snow, 2024).



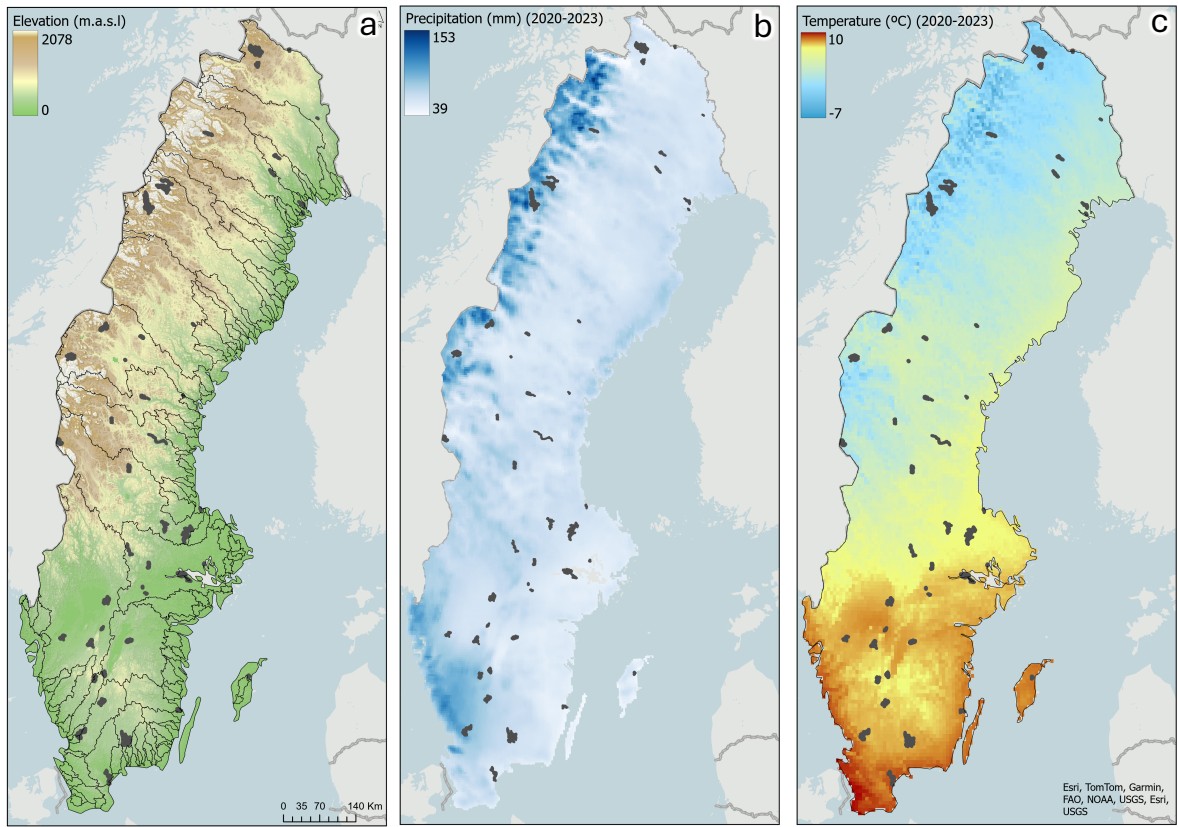

**Figure 1. Spatial distribution of Ramsar wetland study sites (grey polygons) in terms of a) elevation from a 50m resolution DEM by Landmateriet (grey thin lines denote main catchments), b) average precipitation and c) average temperature from 2020 to 2023. Temperature and precipitation data were obtained from the Precipitation Temperature Hydrological Agency's Water Model (PTHBV), available at the Swedish Meteorological and Hydrological Institute (SMHI).**

## 2.2. Wetland characteristics

To place the wetlands into an environmental context, we tabulated each site's latitude, elevation, open water as a percentage of the total area, and general wetland type. The elevation was calculated as the average elevation (m.a.s.l) derived from the Digital Elevation Model 50m (*Markhöjdmodell Nedladdning*, grid 50+) (Lantmateriet, 2022) within the wetland boundary. Open water extent for each wetland was calculated for every month in 2023 using monthly composites of Normalised Difference Water Index (NDWI) binary (water/non-water) masks from Sentinel-2 optical imagery.



The wetland type was estimated using the following databases of wetland classification: (1) The Ramsar Convention database for sites in Sweden, (2) the National Wetland Inventory for Sweden (VMI) (Gunnarsson and Löfroth, 2009), and (3) an updated satellite-based open wetland mapping classification from 2018-2022 (Hahn and

Wester, 2023). Each wetland was assigned a generic wetland class adapted from Gunnarsson and Löfroth (2014): 'open', 'limnic', 'mixed' or 'mire'. 'Open' refers to meadows, grasslands, and temporarily flooded land, 'limnic' refers to lake shores, beaches by watercourses, overgrown lakes and limnogeneous beach complexes. 'Mixed' wetlands are regarded as a combination of multiple wetland types and may include different mires with open or limnic wetland environments. A 'mire' wetland consists primarily of bogs and fens. A fifth wetland type, '*fjäll*' (mountain),

was assigned to wetlands located in Sweden's mountainous regions as they are not classified in the datasets.

## 2.3. Hydrological regime given by water extent analysis

We estimated the hydrological regime from water extent using an automated approach based on remote sensing data. Automatic surface water detection was done with a deep-learning image segmentation model called DeepAqua (Peña et al., 2024). DeepAqua is a self-supervised model with the principal function of detecting surface

water extent in wetlands from Sentinel-1 SAR and Sentinel-2 optical imagery. DeepAqua can detect both open and 'hidden' water using the C-band SAR sensor onboard Sentinel-1, which can penetrate some types of perennial vegetation and detect water due to its emission of longer wavelength (5.6 cm) (Adeli et al., 2021). Usually, semantic segmentation models require manually labelled images as their training label output. With DeepAqua, however, the training labels are generated automatically as NDWI masks (water/non-water) from cloud-free

Sentinel-2 optical imagery of the same location and time as the input training data (SAR imagery). When tested on wetlands in Sweden (Peña et al., 2024), DeepAqua outperformed existing land classification models such as Dynamic World (Brown et al., 2022) and thresholding techniques such as Otsu (Otsu, 1979) on multiple evaluation metrics. We used a pretrained version of the model, which used Sentinel-1 SAR imagery as the training data and NDWI masks derived from the optical Sentinel-2 as the training labels from the same spatial extent and date.


The output predictions comprised polygonised binary water/non-water images for every Sentinel-1 image available between 2020-2023 cropped to within the boundaries of each wetland. The total water area for each image was calculated based on the WGS84 UTM Zone 33N projection. The monthly average of water extent between 2020-2023 was calculated to reduce the risk of annual variability affecting potential clustering while aiming to detect

hydrological regimes under 'average' conditions. Due to extensive snow and ice cover complicating the water extent predictions, the winter months (November, December, January, and February) were removed from the



hydrological regime analysis. We also validated our water extent predictions from DeepAqua in the wetlands with in-situ discharge data available from nearby upstream discharge monitoring stations.

## 2.4. Cluster Analysis

The hydrological regimes obtained through DeepAqua (Section 2.4) were clustered based on their hydrological similarity using a multivariate K-means cluster analysis technique. K-means clustering is a widely used and simple unsupervised machine learning technique in which groups are identified based on the Euclidean distance between a data point and a centroid (a mean of the data) (Everitt et al., 2011). In order to conduct a cluster analysis, data points that characterise the hydrological regime given by water extent are required. We calculated several

hydrological parameters based on each hydrological regime and used them as the input data points (Table A1). The hydrological parameters included known hydrological signatures (Olden and Poff, 2003) and custom parameters to describe the hydrological regime in terms of duration, timing, frequency, magnitude, and rate of change. The optimal number of clusters (k) was chosen based on the silhouette score, which measures the closeness of data points belonging to one cluster to data points of another cluster. The highest value (between -1 and 1) is

then interpreted as the optimal number of clusters. The best-performing parameters were picked using visual inspection (inspecting their ability to cluster the regimes) and validated against collinearity using the Variance Inflation Factor (VIF). The VIF measures the multicollinearity between all variables where <10 is deemed a low level of collinearity.

The emerging pattern given by the silhouette scores indicated that individual hydrological regimes among wetlands were best grouped when $k = 6$-7. Upon visual inspection, $k = 6$ was chosen as the best possible distribution of wetlands into roughly equal-sized groups. The number of sites in each cluster ranged between 4 and 12. Each hydrological parameter was tested individually and in combination with other parameters to see how effectively they helped cluster the wetlands. Certain variables, such as the maximum month, dominated the clustering over

other indices and some indices-pairs were extremely collinear, such as maximum month and minimum month, or Spring/Summer slope difference and slope variation. Therefore, these pairs could not be used together for the final clustering analysis.



# 3 Results and Analysis

## 3.1. Cluster Analysis

We found that from all parameters assessed (Table A1), skewness, kurtosis, normalised maximum slope, number of peaks and baseline month fraction (see graphic depiction in Fig. 2) worked together to form to capture the characteristics of a typical hydrological regime. Upon visual inspection, regimes with similar shapes were grouped together while also maintaining the desired VIF condition (<10) with values of 5.96, 1.84, 3.27, 4.40, and 7.74 for skewness, kurtosis, maximum slope, number of peaks and baseline month fraction, respectively, indicating a reasonable level of non-collinearity. The chosen parameter combination was able to cluster closely related hydrological regimes into six different archetypes named 'autumn drying' (n=12), 'summer dry' (n=10), 'spring surging' (n=4), 'summer flooded' (n=5), 'spring flooded' (n=7) and 'slow drying' (n=5). Support for the archetype names is given by the hydrological parameter results which have been averaged by archetype (Table 2) and are described in section 3.2.

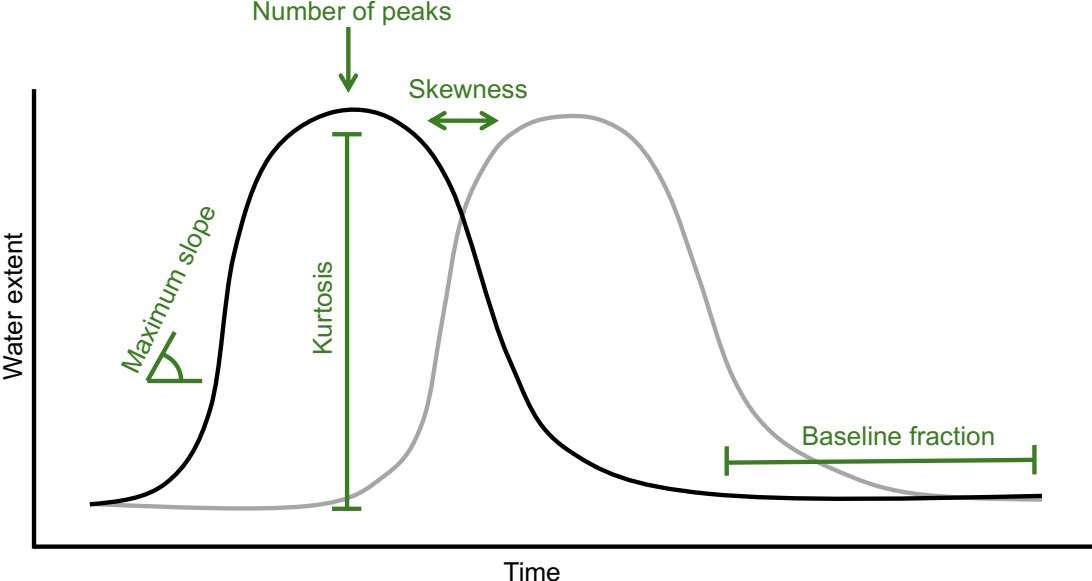

**Figure 2. Graphical representation of the five selected hydrological parameters used to describe the characteristics of the hydrological regime for the final cluster analysis. The parameters include skewness (timing), kurtosis (magnitude), maximum slope (rate of change), number of peaks (frequency), and baseline fraction (duration).**






**Table 1. Overview of the chosen parameter (unitless) combination used for the final cluster analysis of the hydrological regimes given by water extent. Each hydrological parameter represents a unique characteristic of seasonal wetland water extent, which when combined, results in a quantitative description of the hydrological regime. The parameter results are averaged by archetype and the VIF value for each parameter demonstrates a reasonable level of non-**
**collinearity (<10).**

| Hydrological parameter | Interpretation | Autumn Drying | Summer Dry | Spring Surging | Summer Flooded | Spring Flooded | Slow drying | VIF value |
|---|---|---|---|---|---|---|---|---|
| Skewness | Positive skew: wet in early Spring Negative skew: wet in Spring and Summer | -0.13 | 1.83 | 2.00 | -0.60 | 1.18 | 0.40 | 5.96 |
| Kurtosis | High kurtosis: high peakedness Low kurtosis: low peakedness | -1.62 | 2.99 | 4.44 | 1.24 | -0.10 | -0.94 | 1.84 |
| Normalised Maximum Slope | High: fast rate of wetting Low: slower rate of wetting | 0.29 | 0.10 | 0.76 | 0.81 | 0.23 | 0.10 | 3.27 |
| Number of Peaks | <1: no distinguishable flooded month >1: at least one flooded month | 1 | 0 | 1 | 1.2 | 1.14 | 0 | 4.40 |
| Baseline month fraction | High: many months under 'dry' conditions Low: few months under 'dry' conditions | 0.37 | 0.71 | 0.69 | 0.23 | 0.64 | 0.43 | 7.74 |

### 3.2. Hydrological archetype analysis

The overall spatial distribution of the archetypes and thematic graphic descriptions of the general hydrological
regime given by water extent are presented in Fig. 3. *Autumn-drying wetlands* (Fig. 3a) encompass sites with a relatively large water extent from March to July/August, from which drying occurs after that. This archetype is predominantly for wetlands in central and southern Sweden. *Summer-dry wetlands* (Fig. 3b) exhibit the maximum wetland extent at the beginning of the spring, preceding generally dry conditions until October. *Spring-surging wetlands* (Fig. 3c) are only found in northern Sweden and have flashy hydrological regimes. Aside from a dry
baseline condition, they have a brief, one-month period of increased water extent. *Summer-flooded wetlands* (Fig. 3d) remain inundated from May to October after a rapid wetting period and like spring-surging wetlands, they are





only present in the north. *Spring-flooded wetlands* (Fig. 3e) have sites throughout Sweden, although preferentially found in the far north. The hydrological regime of these wetlands resembles that of autumn-drying wetlands, although their drying period occurs earlier in the year. Lastly, central Sweden's *slow-drying wetlands* (Fig. 3f)

exhibit steadily decreasing water extent throughout the summer, reaching minimum water extent in autumn.



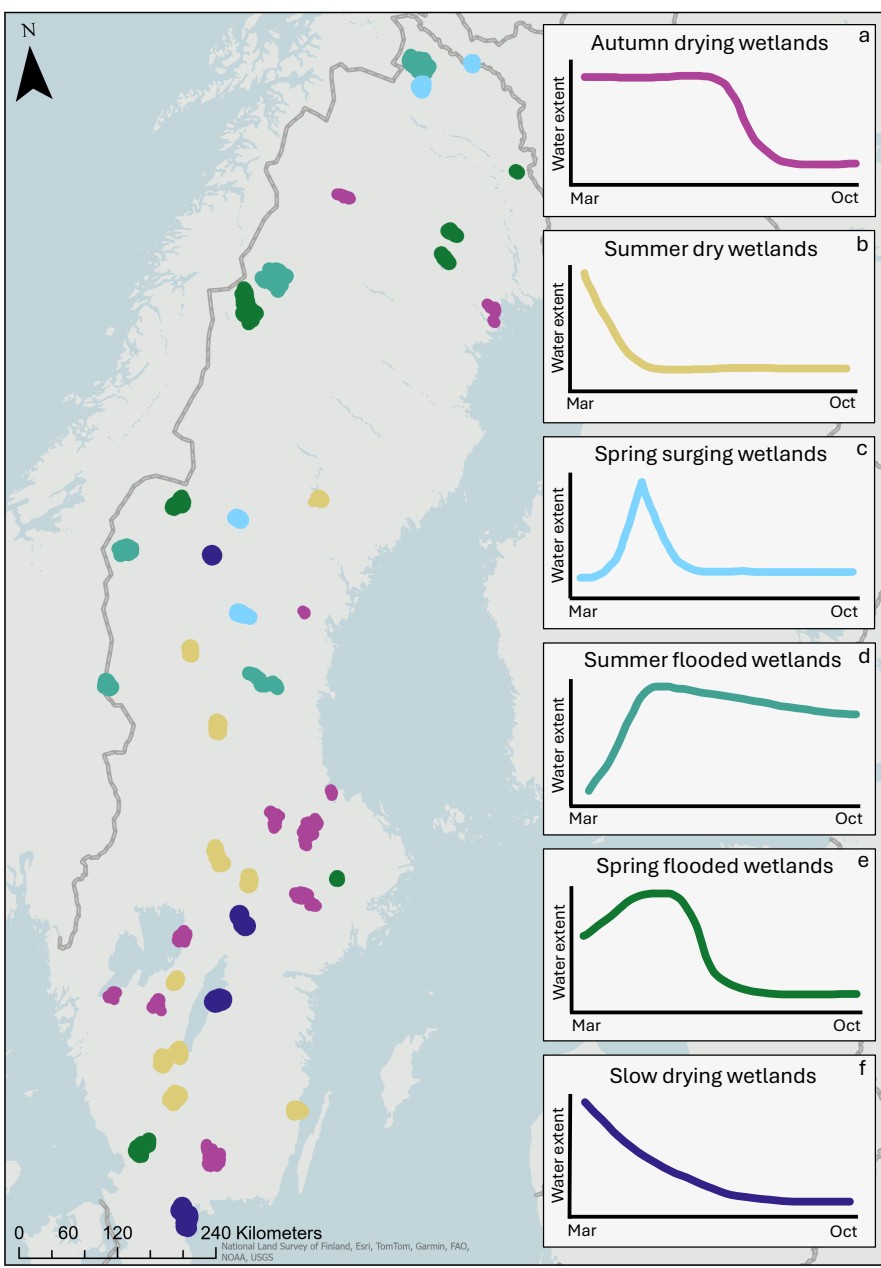

**Figure 3. Spatial distribution of hydrological archetypes for sampled Ramsar wetlands in Sweden (n=43) and representation of their hydrolo9gic regime through the ice-free season (March-October); a – autumn-drying wetlands (n=12), b – summer-dry wetlands (n=10), c – spring-surging wetlands (n=4), d – summer-flooded wetlands (n=5), e – spring-flooded wetlands (n=7) and slow drying wetlands (n=5).**






One of the most distinctive differences between archetypes is the magnitude of water extent at the beginning of Spring. For instance, autumn-drying, summer-dry, and slow-drying archetypes already have large water extents in March and, therefore, do not undergo a rapidly inundating period during Spring or Summer. The lack of any inundation period is reflected in the normalised maximum slope values, which are the lowest out of all archetypes,

suggesting smaller changes in water extent across the year (Table 1; 0.29, 0.10, and 0.10 for autumn drying, summer-dry, and slow-drying, respectively). Additionally, archetypes with large water extent in Spring tend to be found in central and southern Sweden. On the other hand, archetypes such as spring-surging and summer-flooded wetlands start with a small water extent in March which precedes a rapid inundation period. These archetypes, with higher normalised maximum slope values of 0.76 and 0.8, respectively, are more abundant in the north.


A second defining feature between different archetypes is the duration of the 'dry period', defined by months with water extent within the $25^{th}$ percentile of the range. Archetypes with a significant dry period, such as summer-dry and spring-surging wetlands, have both high baseline fractions (0.71 and 0.69, respectively) and positive skewness (1.83 and 2.00, respectively), which indicates that wet conditions are limited to the Spring months. Conversely,

with a negative skewness and low baseline month fraction (-0.60 and 0.23, respectively), summer-flooded wetlands are the only archetype that retains its large water extent throughout the year.

It is worth noting that there is an evident distinction between archetypes with 'peaky' and 'smooth' hydrological regimes. Here we define 'peaky' as hydrological regimes with large variations in water extent, while smooth

archetypes follow a more consistent pattern of monthly water extent changes, which is illustrated in the hydrological parameter results shown in Fig. 4. Peaky archetypes, including spring-surging (Fig. 4c) and summer-flooded wetlands (Fig. 4d), exhibit relatively high values of kurtosis (4.44 and 1.24, respectively), maximum slope (0.76 and 0.81, respectively), and the number of peaks (1 and 1.2, respectively). This is reflected in polar plots that are shifted towards the left hemisphere and occupy a larger total area. On the other hand, 'smooth' archetypes, such

as summer-dry and slow-drying wetlands (Fig. 4f), have polar plots either shifted towards the right hemisphere or occupying a smaller total area. Autumn-drying wetlands (Fig. 4a) do not conform strictly to either 'peaky' or 'smooth' classifications, as they display a mixture of traits that do not align with neither peaky nor smooth. Although we refer to peaky archetypes here, it is important to note that the number of peaks is not necessarily descriptive of just peakeness (kurtosis). For instance, summer-dry wetlands have high kurtosis (2.99) yet zero

peaks, indicating that although they experience large variability in water extent, there is no distinguishable wet month.





**Figure 4. Polar plots for kurtosis, skewness, baseline fraction, number of peaks, and maximum slope for each archetype. Each parameter was initially calculated for each site, and then the average value was computed across all wetlands within each archetype. All parameters are unitless.**





The distinction between 'peaky' and 'smooth' archetypes is further supported by the monthly water extent relative to March of individual wetlands (Fig. 5). Wetlands within the spring-surging, summer-flooded, and spring-flooded archetypes (Fig. 5c-e) demonstrate a distinct peak in water extent between spring and summer. In contrast, the

hydrological regime of summer-dry and slow-drying wetlands (Fig. 5b, 5f) shows a smoother pattern regarding month-to-month water extent variability across all sites. No consistent pattern emerges for autumn-drying wetlands (Fig. 5a), as the archetype exhibits significant variability across different wetlands.







**Figure 5. Hydrological regimes based on water extent for individual and grouped by archetype. The water extent area for each month is shown relative to the water extent area in March. Monthly water extent is an average of all available Sentinel-1 SAR image predictions within each month. Winter months (November to March) are excluded from the hydrological regimes due to snow and ice complicating the water extent predictions. Anomalous wetlands are indicated with an asterisk (*) for Gammelstadsviken and a dagger (†) for Tönnersjöheden-Årshultsmyren.**



Another approach to interpreting archetypes is categorising them based on the breadth of habitats in which they
occur and whether they are multihabitat or habitat-specific archetypes. This classification emphasises that wetlands
can exhibit similar hydrological regimes despite differences in type or location, facilitating a more nuanced
understanding of their functions and services. For instance, autumn-drying wetlands can be classified as a
multihabitat archetype as they span across the entire latitudinal range of Sweden (Fig. 6). On the other hand,
summer-dry wetlands may be considered a habitat-specific archetype as 80% of its wetlands are classified as mires
(Fig. 6d). Another example of a habitat-specific archetype is slow-drying wetlands, which are predominantly found
at low latitudes (Fig. 6a) and elevations with minimal open water (~11%; Fig. 6c). The slow-drying archetype is
also primarily composed of wetlands classified as either open or limnic (Fig. 6d).

Grouping wetlands into archetypes reveals a remarkable similarity in the timing of key features of their
hydrological regimes. Despite the significant variability, 83% of autumn-drying wetlands experience a reduction
in water extent between July and August (Fig. 5a). Similarly, most spring-flooded wetlands reach low water extent
by June, despite varying increases between March and May. This indicates that while hydrological parameters
define archetypes, timing characteristics are also unintentionally captured. Finally, despite many similarities
between wetlands within archetypes, not all wetlands perfectly fit their assigned archetypes. For example,
*Gammelstadsviken* deviates from typical 'autumn-drying' behaviour, with its water extent decreasing earlier in the
year (Fig. 3a). Similarly, *Tönnersjöheden-Årshultsmyren*, classified as a spring-flooded wetland, shows a reduction
in water extent during spring, unlike others in its archetype (Fig. 3e). A possible reason for this is that these
wetlands have unique hydrological regimes that do not conform well with any other wetlands in the dataset, or the
hydrological parameter results did not capture the hydrological regime well in these cases.



**Figure 6. (a-c) Distribution of selected wetland characteristics according to archetype. The boxes represent the interquartile range (IQR), with orange lines indicating the mean value across all wetlands within each archetype. Whiskers outline the full range, and small black circles denote wetlands with anomalous results compared to the rest of the archetype. (d) Stacked bar plot showing the occurrence of wetland types (*fjäll*, limnic, mire, mixed or open) per cluster as a percentage of the total number of sites in each archetype.**



When comparing the hydrological regimes to in-situ discharge data, we found only five wetlands that have an active discharge station located within a reasonable distance upstream (Fig. 7). We validate the hydrological regimes for these wetlands to improve our confidence in the water extent predictions. For wetlands (*Emån*; Fig.

7a-b and Helge å; Fig. 7e-f) with discharge stations very near to (~0.5 km) or within the wetland's boundaries, the average discharge between 2020-2023 agrees well with the water extent data (MSE 0.01 and 0.02, respectively). As the distance between the discharge station and the wetland increases, the level of agreement between the two inevitably decreases. This is evident in the case of *Maanavuoma* wetland (Fig. 7c-d), where the discharge station is located ~17 km upstream. Although the general shape of the hydrological regime of water extent and monthly

discharge are similar, the spring flooding peak of the latter (two months) is more extended than the former (one month) (MSE 0.39). The similarity between the hydrological regime and the monthly discharge was lower (MSE 0.42) for a larger wetland with a high degree of hydrological connectivity such as *Färnebofjärden* (16,866 ha; Fig. 7i-j).







**Figure 7. Left panel: Comparison of hydrological regimes for *Emån*, *Maanavuoma*, *Helge å*, *Östen* and *Färnebofjärden* wetlands based on water extent data between 2020-2023 (black lines) with monthly discharge data averaged from 2020 to 2023 for active on-site or nearby upstream stations (coloured lines) Station ID from SMHI is given in the top left corner. Right panel: Wetland boundaries as defined by the Ramsar Convention coloured by archetype (yellow – summer-dry, blue – spring surging, purple – slow drying, pink – autumn drying). The location of the discharge stations used for data is marked with black and white circles, and watercourses are shown in dark blue. Light blue background depicts water, and grey indicates land in the basemap.**

## 4 Discussion

### 4.1. Discussion of results

One of the defining features for most archetypes was the timing of pronounced water extent change, which only became apparent when the sites were grouped. Therefore, we emphasise the usefulness of employing archetypes in hydrological studies, as hydrological regimes may not be best evaluated across sites when using a single parameter.

Overall, we define the following six major archetypes:

1. *Autumn-drying wetlands.* Limnic, open or mixed wetlands found across Sweden with a high proportion of open water undergoing prolonged wet periods that precede rapid drying between July and August.

2. *Summer-dry wetlands.* Predominantly mire wetlands preferentially found in central and southern Sweden with large water extent in early spring followed by a prolonged dry period.

3. *Spring-surging wetlands.* Peaky wetlands in the north of Sweden preferentially dry except for a one-month surge in water extent in May.

4. *Summer-flooded wetlands.* Mire and fjäll wetlands found in northern, mountainous Sweden experiencing rapid inundation period between spring and summer with a prolonged wet period.

5. *Spring-flooded wetlands.* Mire and fjäll wetlands found mainly in northern Sweden with a wet period during the Spring that precede a prolonged dry period beginning in June.

6. *Slow-drying wetlands.* Open or limnic low wetlands in the south with no wetting period between March and October and a slow drying throughout the year.

While classifying wetlands based on hydrological regimes does not account for vegetation, soil type, or climate, we propose that the hydrological regime serves as a proxy for the cumulative effect of these characteristics. In fact, Bullock and Acreman (2003) conclude that grouping wetlands based on their local wetland classification term is less intuitive than grouping them based on their hydrological types. Moreover, multihabitat and specific-habitat





archetypes in our study indicate that wetlands can share similar hydrological regimes despite different environmental conditions. Therefore, there is value in using the hydrological regime to understand ecosystem services in wetlands better, providing that it is also complimented with other environmental data (Poff et al., 1997).


We suggest that there are two main reasons why some wetlands like *Gammelstadsviken* and *Tönnersjöheden-Årshultsmyren* were not easily categorised. Firstly, the indistinct nature of some wetlands suggests that some hydrological regimes can sometimes be seen as a continuum rather than easily separated categories, making it challenging to group them into distinct archetypes. Secondly, the limited scope of the wetland database used for
clustering might have excluded the existence of additional archetypes that can be obtained when focusing on the hydrological regime from water extent changes. It is also important to note that while we defined archetypes using an average of four years of monthly water extent data, these may only reflect the observed period. Longer observational periods are necessary for determining extended trends and the impact of changing climatological conditions.


Although detailed exploration of the drivers of the hydrological regime is beyond this study's scope, we theorise that different hydrological regimes in wetlands may partly stem from hydrological factors such as the wetland area to watershed area ratio, watershed location, snow melt upstream and surface connectivity. For example, spring-surging wetlands, with few surface water inlets, rely mainly on snowmelt, remaining dry for much of the year. In
contrast, summer-flooded wetlands have a larger supply of water from the inflow of multiple streams and remain inundated longer (Lane et al., 2018). Smaller watershed-to-wetland area ratios such as those found for spring-flooded wetlands lead to quick peaks and rapid declines in water levels, while larger ratios (like slow-drying wetlands) result in lower but more prolonged flood peaks (Davie and Wyndham Quinn, 2019). Additionally, wetlands in headwaters, like spring-surging and summer-flooded wetlands, experience rapid flood peaks
characteristics of upper catchment water flows (Morley et al., 2011).

## 4.2. Discussion of methods

Using water extent as our key measurement, SAR imagery provided data with dense spatiotemporal resolution across 43 wetland sites, which in theory can be applied to any wetland larger than 200 ha. The reliance on remote
sensing is driven by a lack of in-situ data, which would have partly or wholly missed the hydrological regime signatures for most of the chosen wetlands in this study.



In order to make use of the abundance of remotely sensed data, we chose the automatic deep learning-based approach (DeepAqua) to detect water extent since the model can predict instances of surface water using SAR imagery for hundreds of images without any manual annotation. SAR is particularly useful for wetland studies since most water goes undetected by optical imagery (Sahour et al., 2021). However, it is possible that C-band radar can underestimate surface water extent in wetlands since its relatively shorter wavelength limits the penetration capacity into denser vegetation (Adeli et al., 2021). Additionally, we assume that surface water extent is analogous to total water storage, which may not be true for mire types where water is predominantly stored within the soil (Acreman and Holden, 2013) or for wetlands constrained by vertical landscape features which may lead to the assumption that the water in the wetland remains constant without considering water level. Therefore, although water extent is a useful descriptor of hydrological regime, it may be beneficial to include water level data, such as from the new Surface Water and Ocean Topography (SWOT) mission that launched in 2023 (Hamoudzadeh et al., 2024). The main caveat of all existing remote sensing-based methods is the influence of snow and ice during the winter months affecting the backscattering signal of the radar sensor. Water extent data for the winter months remain a crucial element for fully understanding hydrological regimes. For instance, many of the presented archetypes have a small water extent in October, which introduces the question of how the wetland is recharged again to reach its relatively high water extent after winter. In-situ discharge data could be used to fill the data between for the winter months, but this is currently challenging since there are so few active stations nearby the observed wetlands.

As well as the lack of discharge stations, discharge data could also not be used to validate the hydrological regime for wetlands with multiple inlets (e.g. *Farnebofjärden* wetland, Fig. 7i-j), since all inlets likely contribute to the overall hydrological regime of the wetland and may deviate from the seasonal discharge pattern observed at the station. A more comprehensive in-situ station network should be installed closer to or on-site of wetlands of interest for future validation efforts.

### 4.3. Hydrological regimes for hydrological functions

In this paper, we quantified the hydrological regimes for wetlands in Sweden to better understand their hydrological functions. Information on hydrological functions can therefore indicate which ecosystem services are relevant for that wetland. We find that archetypes such as spring-flooded wetlands and spring-surging wetlands, which have high water extent during the spring and low water extent during the summer, are akin to headwater



wetlands. Headwater wetlands are known to increase flood flows during the wet season while decreasing dry season flows (Bullock and Acreman, 2003). Therefore, we suppose that wetlands belonging to these archetypes do not provide flood control as a prominent ecosystem service, but rather the opposite; they tend to exacerbate
flooding (Åhlén et al., 2022). However, evidence suggests that headwater wetlands can temporarily store immediate floodwaters (Kadykalo and Findlay, 2016), although more data is required to investigate the lag time between wetland storage and downstream discharge for our archetypes.

The converse appears to be true for autumn-drying wetlands and slow-drying wetlands, which have characteristics
of floodplain wetlands. There is substantial evidence (Acreman and Holden, 2013; Golden et al., 2021) to suggest that floodplain wetlands reduce or delay floods, which may be shown in the latency of drying throughout the season for autumn-drying wetlands and slow-drying wetlands. These archetypes store the water for longer periods, suggesting that they simultaneously reduce flood peaks and increase water flow during the dry summer. Although we did not do a detailed analysis of ecosystem service delivery or have data from downstream discharge stations
to confirm flood attenuation capacity (Andersson, 2012), this work provides a starting point for identifying potential future Ramsar sites or areas to prioritise for protection and management in regions with prominent flooding and summer drought.

Another important asset of hydrological regime studies is the ability to determine hydrological functions at any
given time, as these functions can shift depending on the wetland's state (Spence et al., 2011). For instance, water extent variability can indicate the timing of the threshold between storage and runoff (Yanfeng and Guangxin, 2021). Flashy water extent variability in northern archetypes like spring-surging and summer-flooded wetlands suggests a switch to conditions where water is not stored but rather flows straight through the wetland downstream. This shift may result from frozen ground or sporadic permafrost hindering water storage in soils (Yanfeng and
Guangxin, 2021) and/or spring snowmelt contributing to over half of the annual flow in a short period (Spence et al., 2011).

Aside from hydrological-related ecosystem services, it is important not to overlook other archetypes that may offer other valuable ecosystem services, such as maintaining biodiversity and carbon sequestration. For instance,
biodiversity is driven by the wetland hydrological regime due to variations in water tolerances among vegetation species. Additionally, wetlands classified under the 'northern' archetypes are particularly significant for carbon




sequestration. Future research that differentiates between hydrological regimes present in carbon-sequestering wetlands can further improve our understanding of their ecosystem services (Kirpotin et al., 2011).

## 5 Conclusion

This research aimed to improve our understanding of wetlands by revealing their hydrological regimes using remotely sensed data on water extent. We chose an automatic detection method based on Sentinel-1 SAR imagery because it can operate in cloudy and dark conditions and sometimes detect water under vegetation. The hydrological regimes were grouped based on similar hydrological characteristics identified by custom hydrological parameters. For 43 Ramsar sites in Sweden, the hydrological regimes based on monthly water extent between

2020-2023 could be grouped into six distinct archetypes. The defining traits were mainly related to the timing of change and the duration of wet and dry periods. Despite heterogeneity in the archetypes' spatial distribution, flashy archetypes with high water extent variability were preferentially found at higher elevations and latitudes, while less variable and drier archetypes were concentrated towards low elevations and latitudes. Additionally, mire types were more homogeneous and thus more likely to be part of the same archetype compared to open or limnic wetland

types.

While contextual information is vital for our deeper understanding of wetlands, rich data can be drawn from simply tracking the water extent through time, such as insight into runoff and storage dynamics. Furthermore, by reducing multiple wetland hydrological characteristics to the hydrological regime, we demonstrated that we could use the

notion of archetypes to infer information about their specific hydrological functionality nationwide. Since many archetypes consist of multiple wetland classifications, we recommend estimating hydrological functions based on the hydrological regimes, not individual wetland types. By being able to draw information from the archetypes, we reveal a new understanding of the hydrological functioning of wetlands with a particular emphasis on hydrological-related ecosystem services such as flood control and water supply during low flow periods.




**Appendix A**

**Table A1. Hydrological parameters used for cluster analysis. Each parameter was evaluated individually and in combination with others to assess its effectiveness in capturing the characteristics of the hydrological regime. (N) – Normalized to remove the effect of wetland size.**

| Hydrological parameters | Description |
|---|---|
| Max Month | Timing of the highest water extent |
| Min Month | Timing of the lowest water extent |
| Standard Deviation | Measure of dispersion of water extent values in a dataset |
| Skewness | Measure of symmetry in a distribution of water extent values |
| Kurtosis | Measure of peakedness in a distribution of water extent values |
| Coefficient of Variation | Measure of the dispersion water extent values around the mean |
| Range (N) | Difference between the maximum water extent value and the minimum water extent value, normalised to the mean wetland size |
| Minimum slope (N) | Smallest slope of monthly water extent change taken from the first derivative, normalised to the water extent range |
| Maximum slope (N) | Highest slope of monthly water extent change taken from the first derivative and normalised to the water extent range |
| Spring/Summer Area Difference (N) | Difference between the average spring water extent (in March, April and May) and average summer water extent (June, July, August), normalised to the mean wetland size |
| Spring/Summer Slope Difference (N) | Difference between the average spring slope of monthly water extent change (in March, April and May) and average summer slope of monthly water extent change (June, July and August), normalised to the mean wetland size |
| Slope Variation (N) | Standard deviation of all month-to-month slopes of monthly water extent change, normalised to the water extent range |
| Number of Peaks | Number of peaks, defined as a relatively high value of water extent between two relatively low values of water extent |



| Baseline Month Fraction | Number of months within 25th percentile of the distribution of water extent values as a fraction of the year |
|---|---|


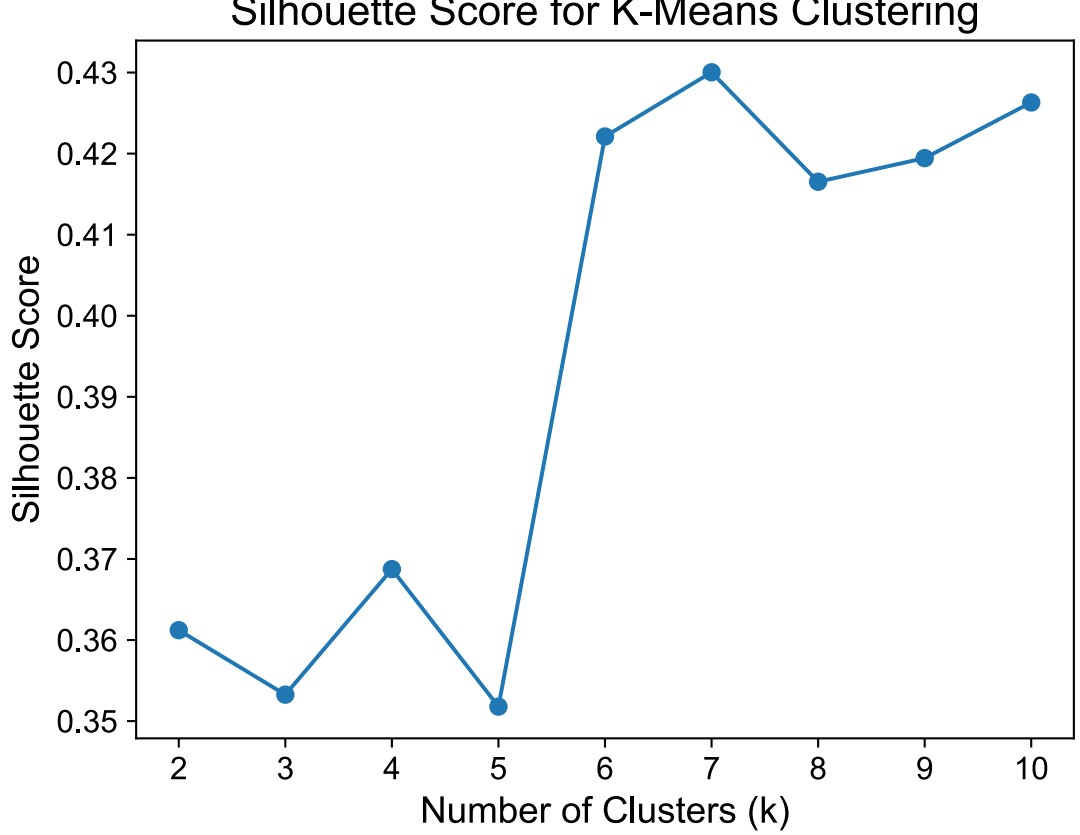

**Figure A1. Silhouette score (range -1 to 1) as a measure of closeness of data points belonging to one cluster to data points of another cluster for k=2-10.**





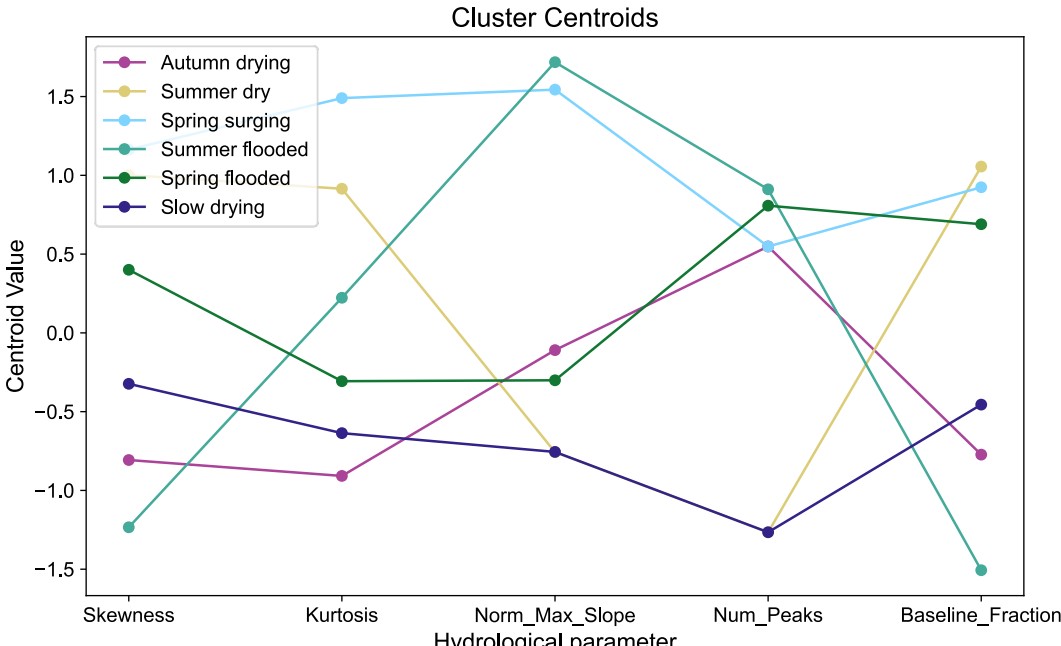

**Figure A2. Centroid points for each of the final hydrological parameters per cluster. Centroid points are defined as the coordinates (Euclidean) of the cluster centres.**



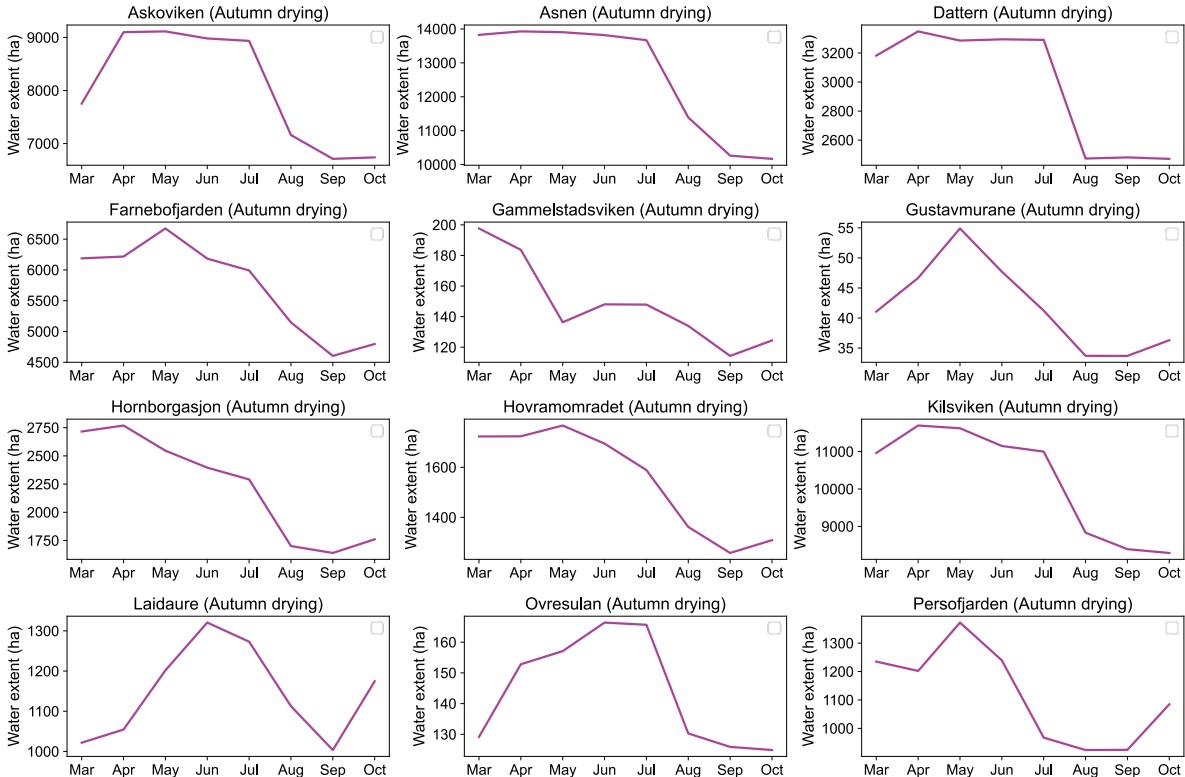

**Figure A3. Average monthly water extent (March-October) between 2020-2023 for all wetlands belonging to the autumn-drying archetype**






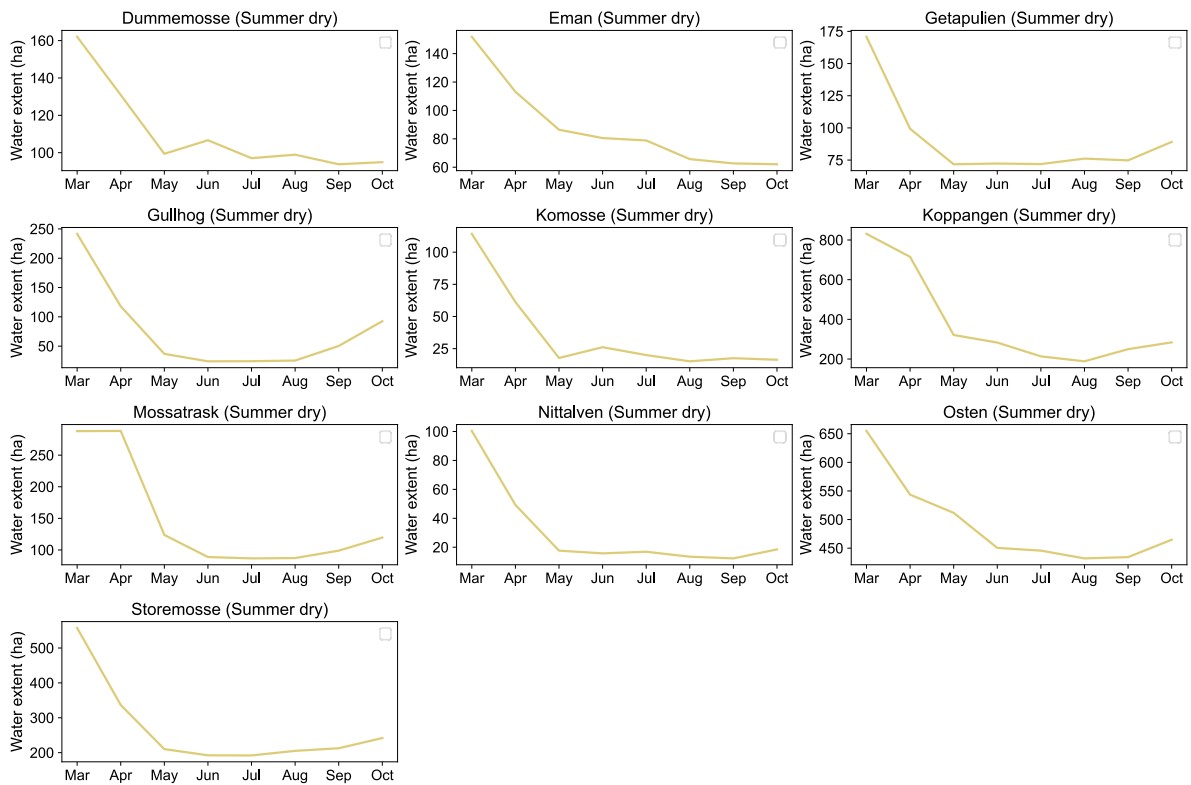

**Figure A4. Average monthly water extent (March-October) between 2020-2023 for all wetlands belonging to the summer-dry archetype**





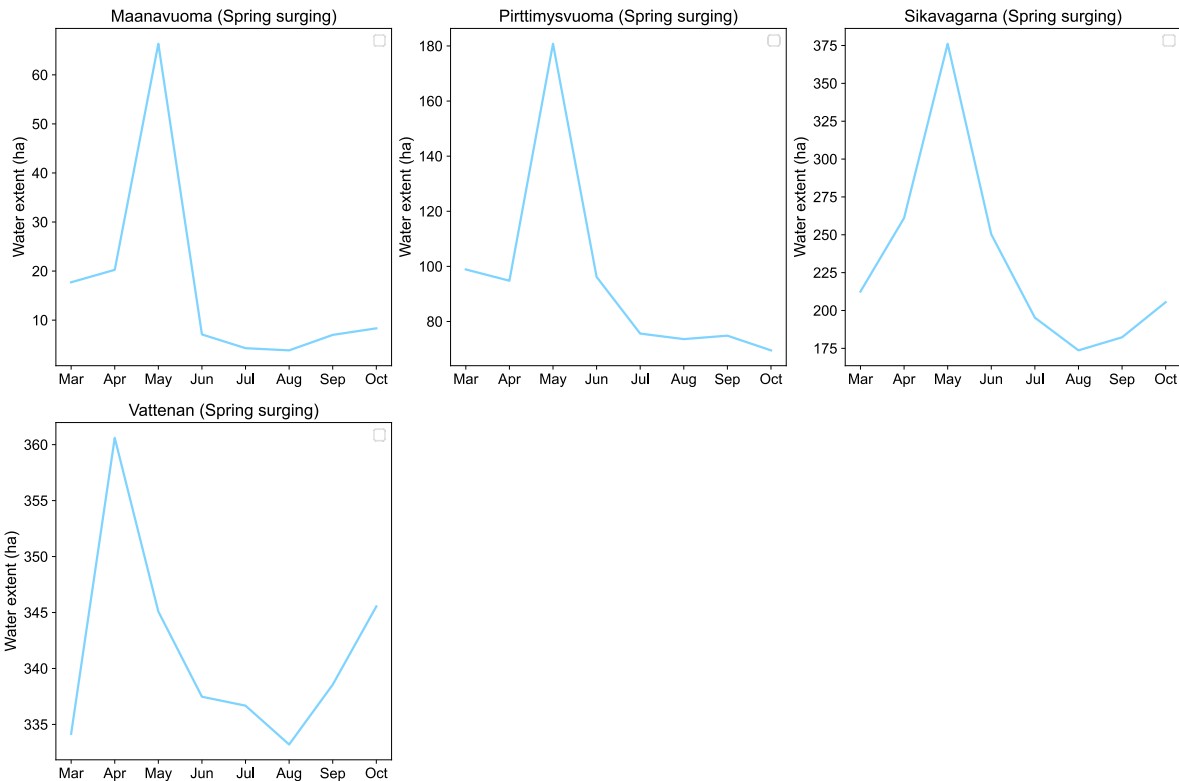

**Figure A5. Average monthly water extent (March-October) between 2020-2023 for all wetlands belonging to the spring-surging archetype**





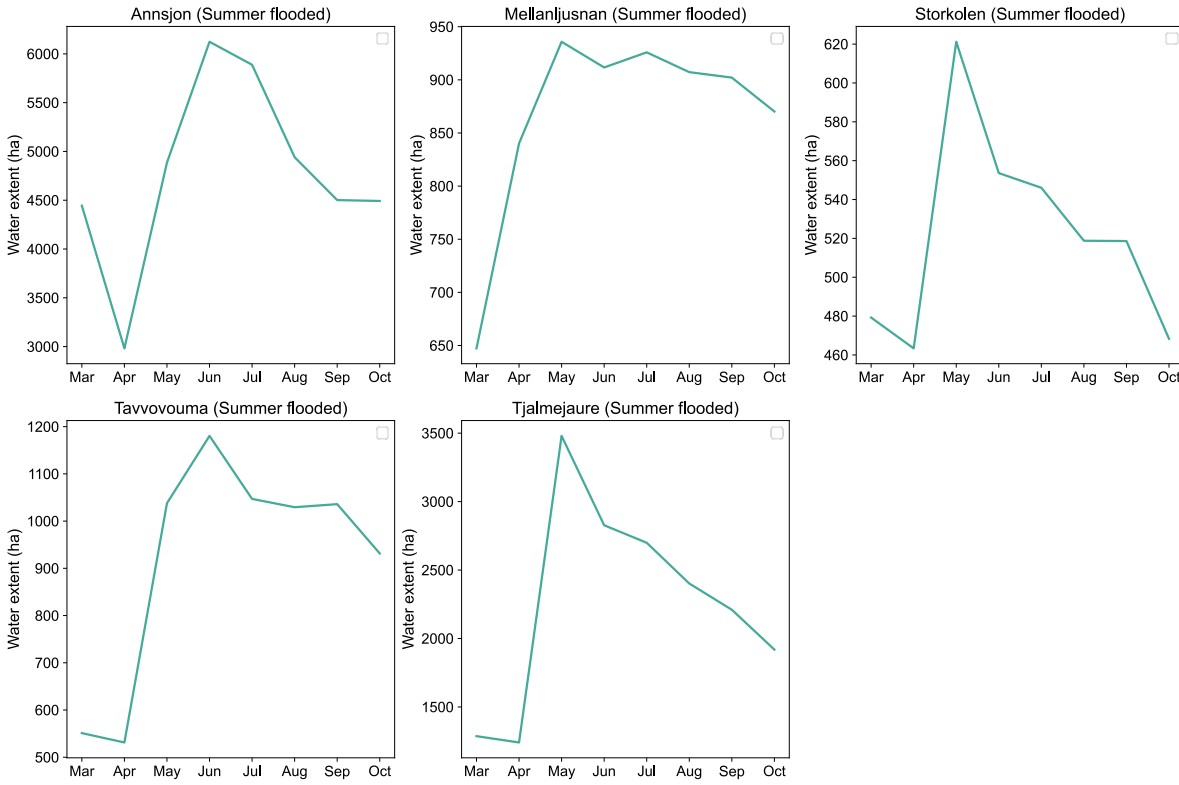

**Figure A6. Average monthly water extent (March-October) between 2020-2023 for all wetlands belonging to the summer-flooded archetype**





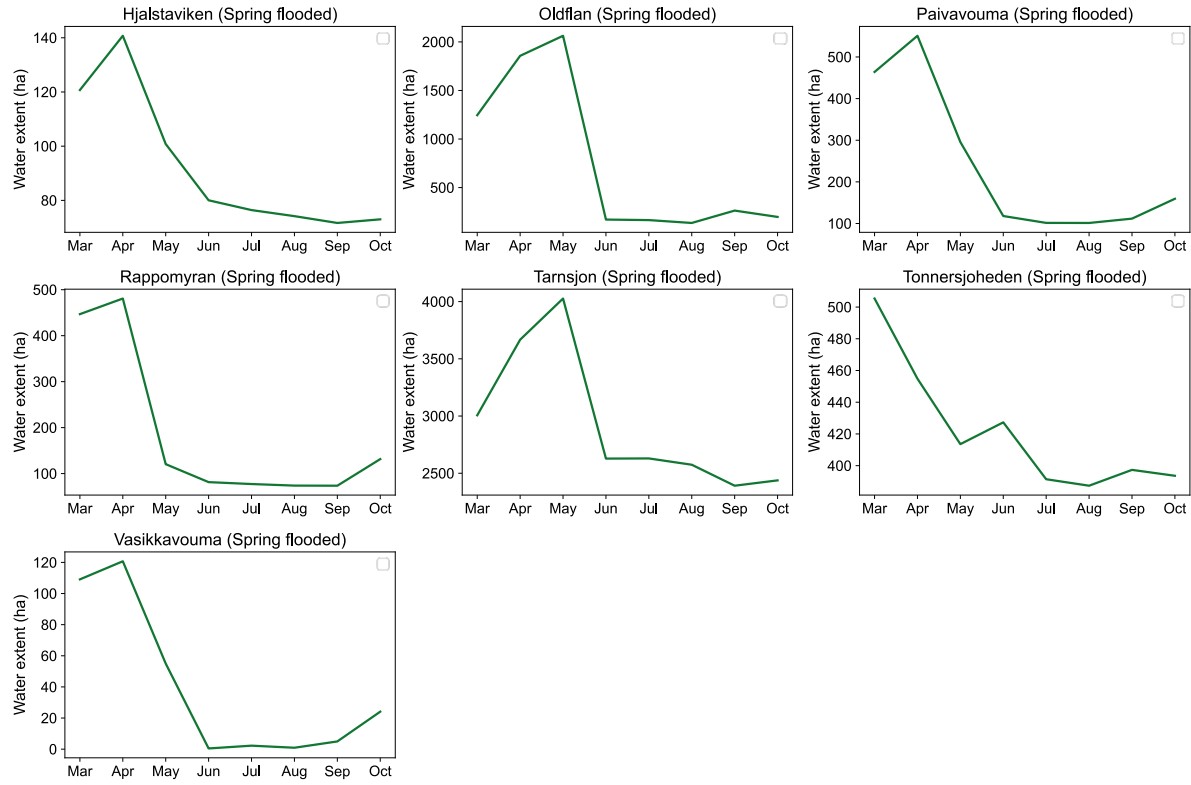

**Figure A7. Average monthly water extent (March-October) between 2020-2023 for all wetlands belonging to the spring-flooded archetype**

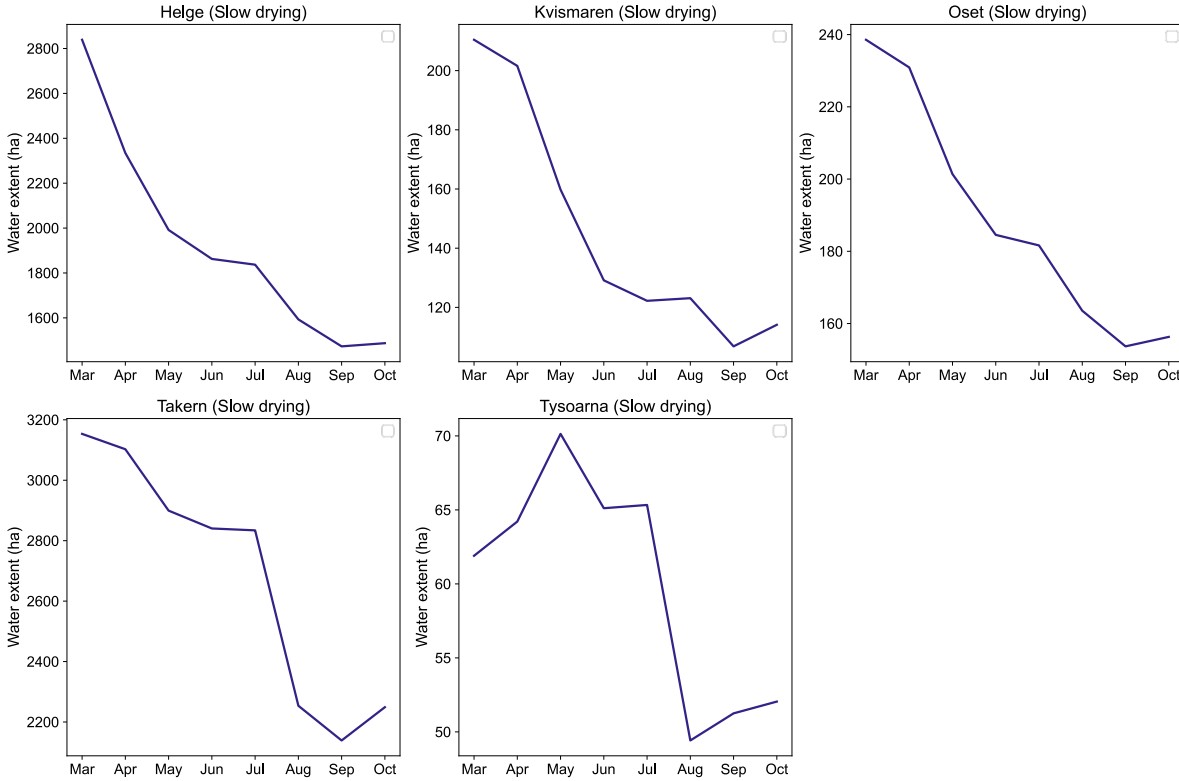

**Figure A8. Average monthly water extent (March-October) between 2020-2023 for all wetlands belonging to the slow-drying archetype**


*Code and data availability.* All data including, environmental data, hydrological parameter results and raw water extent data for all wetlands is available through (Robinson, 2024) (https://doi.org/10.5281/zenodo.13833605). Code for processing data and cluster analysis is available at https://github.com/ab-e-rob/hydrological_archetypes.

Code for predicting water extent in wetlands using DeepAqua can be found at https://github.com/melqkiades/deep-wetlands.

*Author contributions.* AR conceptualised the project, developed the methodology, conducted data collection and analysis. AS and FP contributed to methodology development and provided research guidance. PH assisted in results interpretation. FJ provided guidance on methodology, data analysis, and interpretation; supervised the



project. The co-authors all contributed to the preparation of the paper. The use of ChatGPT helped to improve prose.

*Competing interests.* The authors declare that they have no conflict of interest.

*Financial support.* This project was funded by Svenska Forskningsrådet Formas (Formas) Project 2022-01570.

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
