# Peer review of "The Hydrological Archetypes of Wetlands"

_EGUsphere, 2024_

## Author Response (AR1)

**AC1**

We sincerely thank the Editor, Associate Editor and Reviewers for handling and taking the time to read and review our manuscript. We are also grateful for the reviewer's insightful and detailed comments, and we believe that this work has been greatly improved as a result. Below are our replies to the comments given:

**Introduction**

Comment 1: Line 84-86, how often is Sentinel-1, how often can you cover 43 Ramsar wetlands? Monthly? In one month, how many time revisit the same point?

Response 1: Up until late December 2021, Sentinel-1 had an average revisit time of about 6 days for all 43 wetlands. Wetlands further north have a slightly better temporal resolution, as higher latitudes have higher revisit times. However, due to the failure of Sentinel-1b in December 2021, the revisit time was reduced to around 10-12 days on average for all wetlands, with only Sentinel-1a working and transmitting data in the period 2022-2023. So, depending on the year and wetland, there is data availability for 43 Ramsar wetlands 2-6 times per month.

We have made this more explicit in the manuscript on lines 144-145 by changing the sentence to "We use the case of 43 Ramsar wetlands as they are well inventoried, present good spatiotemporal coverage of SAR data (~1-2 passes per week between 2020-2021, after which spatiotemporal coverage is reduced to ~10-12 days due to the failure of the Sentinel-1b satellite)".

**Methods**

**Comment 2: Line 97-98, how do you define poor SAR data availability?**

Response 2: We mention poor SAR data due to processing issues with the SAR tiles which lead to many corrupted tiles in one month. These tiles have no return signal and therefore no backscattering information. Also, due to the failure of Sentinel-1b, some wetlands had very poor coverage (

Peña et al. (2024) split the SAR/NDWI image pair into 64x64 tiles which resulted in 45,500 image-label pairs. The training-validation datasets were generated by using a conventional 80/20 split, where 80% of the total tiles were used for training, and 20% for validation.

Lastly, the test dataset comprised three Ramsar wetlands located in low-lying areas of central and southern Sweden – Svartådalen, Hjälstaviken, and Hornborgasjön.

We now explicitly mention in the text that no new training was performed for this paper by editing lines 145-148: "We use a **pre-trained** version of the DeepAqua model for our analysis, which was trained on a Sentinel-1 and Sentinel-2 based NDWI binary image over central Sweden from the 5th June 2018."

Comment 6: In methods section, no description about validation for water extent prediction. All the analysis is based on water extent result, validation for it should be described here.

Response 6: Thank you for raising this important point. Since there are no ground-truth data of dynamic wetland water extent in any of the Swedish Ramsar wetlands, we have developed a validation that consists of two steps. The validation includes comparing our water extent predictions with 1) manually annotated water extent and 2) in-situ discharge data Based on your comment, we now mention the validation more explicitly in Methods section 2.3. and discuss the results in Results section 3.1.

The first, manual annotation, was performed to assess the accuracy of water extent predictions from DeepAqua. Although we acknowledge that manual delineation of water extent from SAR imagery is not technically 'ground truth', we wanted to validate the water extents that were predicted by the model using our interpretation of wetland water extent from SAR imagery. We deemed it reasonable to manually annotate wetland water extent for a systematic sample of wetlands (5 in total) for all images available from the year 2021 since manual annotation is very time-consuming. For the manual annotation, we use the Sentinel-1 backscattering data which helps identify both open water and water surfaces below grassy, floating or sometimes bushy vegetation due to the wavelength of the C-band of the Sentinel-1 SAR signal.

We randomly chose one wetland **per archetype** to compare our manual estimates with the DeepAqua predictions to get a representative sample of wetlands. These wetlands were 'Maanavuoma (Spring-surging)', 'Tysöarna (Spring-flooded)', 'Dättern (Summer-flooded)', 'Store

mosse (Slow-drying)' and 'Hjälstaviken (Spring flooded)'. The figure below shows the comparison between the manual estimate vs the DeepAqua estimate. A table of the average mean root-square error (RMSE) and normalised mean root-square error (NMRSE) are available in the Supplementary\_data.xlsx file that accompanies the manuscript, which is published on Zonodo (<a href="https://doi.org/10.5281/zenodo.13833605">https://doi.org/10.5281/zenodo.13833605</a>). We discuss this additional analysis in the manuscript in Results section 3.1.

Figure 1 (Figure 2 in manuscript). (a-e) Comparison between monthly water surface extent from DeepAqua predictions oand manual delineation in 2021. (f) Values of Normalised Root Mean Square Error (NRMSE; RSME divided by the range in wetland extent) between manually delineated and DeepAqua predictions.

The second aspect of the validation involved comparing the time series of upstream and downstream discharge data from the Global Runoff Data Centre (GRDC) and the Swedish Meteorological and Hydrological Institute (SMHI) with that of wetland surface water extent from DeepAqua in matching dates. In the previous version of the manuscript, we had compared the water extent predictions with discharge data from stations upstream of the wetland, and as the reviewer rightly pointed out, we had only 5 of 43 wetlands validated in this manner. In order to improve our validation efforts, we have now included all downstream discharge stations as well, which increases the overall dataset from 5 to 23 wetlands. We also note whether the watercourse between the wetland and the discharge station is regulated and the distance between the wetland and the station. A discussion of the discharge results is now included in the manuscript under Results section 3.1. All plots and RMSE/NRMSE are also available in the Supplementary data Excel file published on Zonodo. The figure below shows the NRMSE between the surface water extent data and the discharge data for the 23 sampled wetlands.

Figure 2 (Figure 3 in manuscript). (a) – NRMSE between daily discharge and wetland water extent for the 23 wetlands with available discharge data. Green boxes indicate the interquartile range (IQR), whiskers represent the full range, and orange lines show the mean NRMSE. (b) Mean NRMSE versus mean discharge for each wetland, calculated over matching dates from January 2020 to August 2023. Wetlands with regulated flow paths between the wetland pour point(s) and discharge station are indicated by black outlines.

We had also performed a third validation approach for the remaining sites that were not validated using manual annotation or discharge data, by comparing our water extent predictions to the Dynamic World land-use land-cover classification dataset published by Brown et al. (2022). The latter is the result of training a deep-learning model based on optical data (A fully convolutional neural network). Brown et al. (2022) report a 94% accuracy for open water and ~42% accuracy for flooded vegetation, which were combined and compared to our DeepAqua predictions as a monthly wetland water estimate between 2020-2023. However, since Dynamic World may not act as an accurate validator of our water predictions in wetlands as the accuracy for flooded vegetation in Dynamic World is quite low, we decided not to include this third validation component in the new version of the manuscript. Yet, we show the reviewers these results as a background check and as part of this response in the Supplementary data Excel file on Zonodo.

Comment 7: In lines 115-116, where is the result of each site's latitude, elevation, open water as a percentage of the total area, and general wetland type? Is it Figure 6? If yes, please cite it in lines 115-116.

Response 7: Thank you for pointing this out, we have now cited Figure 6 on line 119.

**Results and Analysis**

Comment 8: In lines 183-186, what does it mean that VIF values 5.96 for Skewness, can you explain more how does VIF value work? In line 167, you mentioned VIF values measure between all variables. So VIF value 5.96 of skewness measures multicollinearity with all other variables or some variables?

Response 8: The Variance Inflation Factor (VIF) measures the degree of multicollinearity between one independent variable with all the other independent variables. The VIF works by calculating how much the variance of a regression coefficient is increased due to correlation with other independent variables. For example, a value of 5.96 for Skewness implies that the variance of the regression coefficient is inflated by a factor of ~6 compared to what it would be if Skewness were completely uncorrelated with other parameters. We used VIF to statistically show if the five hydrological parameters are strongly correlated as high multicollinearity implies that the hydrological parameters are more likely to describe the same hydrological regime characteristic i.e., no two parameters should describe the magnitude of the hydrological regime, for example.

We have made this interpretation clearer in the text on lines 185-192: "The best-performing parameters were picked using visual inspection (inspecting their ability to cluster the regimes) and validated against multicollinearity using the Variance Inflation Factor (VIF). The VIF measures the degree of multicollinearity of one hydrological parameter with all other parameters by calculating how much the variance of the regression coefficient increases due to correlation with other independent variables. We recognise that there is some degree of inherent correlation between the hydrological parameters since they are descriptors of the same hydrological regime. Therefore, we used a VIF value of

Figure 3 (Figure 5 in manuscript). (a) Overview of the chosen parameter (unitless) combination (averaged by archetype) used for the final cluster analysis of the hydrological regimes given by water extent and the VIF value for each parameter. (b) Graphical representation of the five selected hydrological parameters used to describe the characteristics of the hydrological regime for the final cluster analysis. (c) Radar plots for for final hydrological parameters averaged by archetype.

**Comment 14: Is figure 5 same as figures A3-A8? One is relative water extent, one is absolute water extent?**

Response 14: Yes, this is correct, Figure 5 is relative water extent whereas A3 and A8 are absolute water extents. We decided to include the relative water extent in the main body of the manuscript for two reasons: 1) to emphasise the seasonal evolution of wetland water extent,

and 2) to allow comparison between wetlands with different surface areas. We also wanted to include the absolute water extents as plots in the appendix for any reader that would be interested in a specific site. Note that the water extent plots are now referenced as Figures A2-A6.

Comment 15: In lines 274-282, The main topic of the paper is to classify wetlands based on hydrological regimes, but this paragraph is more about the classification of habitat types. Is this classification closely related to the main topic (hydrological regime)? It is recommended that the authors explain why this classification is necessary and how it contributes to the main line of research.

Response 15: Thank you for raising this point. The section about interpreting archetypes as multihabitat and habitat-specific is indeed more about ecology than the hydrological regime, although the two are intrinsically linked. What we wanted to emphasise in this paragraph is that apparently, some archetypes are more heterogenous (or variable) than others in terms of their hydrological regimes and environmental characteristics. We have removed the terms habitat-specific and multi-habitat archetypes and mention instead hydrological regime heterogeneity between archetypes. We return to this point in the discussion, whereby we outline the importance of hydrological archetypes, since the same wetland type or environment may not produce the same hydrological regime. The paragraph on lines 364-382 is as follows:

"Another approach to interpreting archetypes is by examining the degree of homogeneity within each archetype. This is because some archetypes share more similarities in terms of their environmental characteristics and hydrological regimes. For instance, summer-dry wetlands are mostly comprised of mires or open wetlands (Fig. 8d), typically lying at low elevations and exhibiting similar hydrological regimes (Fig. 7e). Spring-surging wetlands are also considered a homogenous archetype since i) they are located primarily in high-latitude regions (Fig. 8a), ii) are mainly fjäll wetlands, and iii) tend to have little variability in their hydrological regime (Fig. 7a). In contrast, spring-flooded and summer-flooded wetlands are found all over Sweden, across a range of elevations (Fig. 8b) and encompass many different wetland types. This highlights that hydrological regimes are not always associated with a specific wetland type, but rather depend on the broader archetype to which the wetland belongs."

Comment 16: In lines 287-288, hydrological parameters are from water extent timing characteristics. So it is not unintentionally, it is intentionally in the input data.

Response 16: In line 387, we have changed the sentence to "This indicates that the hydrological parameters expectantly capture timing characteristics..."

Comment 17: In figure 7, legend is "water extent" but second y axis label is "wetland extent". The whole manuscript is talking about water extent, please correct it.

Response 17: The coloured polygons in the right panel are the areas defined as Ramsar sites by the Ramsar Convention. We make this clearer by changing the legend to 'Ramsar area' instead of 'wetland area'.

Comment 18: In figure 7 caption, "water extent data between 2020-2023 (black lines) with monthly discharge data averaged from 2020 to 2023 for active on-site or nearby upstream stations (coloured lines)", it seems "black lines" and "coloured lines" should be swapped or the line color in figures should be swapped.

Response 18: We have now swapped the labels around in the caption for Figure 7 which is Figure 4 in the new version of the manuscript, so that the coloured lines are for water extent and black lines are for discharge.

Comment 19: In lines 302-321, you validated 5 wetlands due to in-situ data limit, but in total there are 43 Ramsar sites are analyzed, for other 38 sites, should at least compare the water extent with other water extent datasets to validate its accuracy. In addition, the whole manuscript is based on the water extent result, so the validation part should be shown in the beginning of Section 3, instead of the last paragraph.

Response 19: As we explain in Response 6, we have now extended our validation of the hydrological regime by comparing 1) the DeepAqua predictions with our manual estimates of water surface extent and 2) against on-site discharge data in a larger set of wetlands, including operating discharge stations that are not only upstream of the wetland, but downstream too. This extends the original validation dataset from 5 to 26 wetlands, with an additional background check for the rest of the wetlands based on comparison with the Dynamic World dataset.

We agree with the point of view of the reviewer; it feels more natural to put the validation at the beginning of the results. We also believe that having the validation at the end of the results somewhat dilutes the message about hydrological archetypes. Therefore, we accepted the suggestion of the reviewer and moved the validation section to the beginning of the results in Section '3.1. Surface water extent validation'. By moving the validation to the beginning of the results, we also believe that it gives more strength to results since they are validated before they are discussed, so we are grateful for the suggestion.

Comment 20: In lines 306 and 311, please tell the full name of "MSE".

Response 20: For our validation, we changed the error metric to the Normalised Root Mean Square Error (NRMSE) instead of MSE, and we define the metric in the methods on line 169.

**Discussion**

Comment 21: Lines 330-341, seems a repetition of lines 210-220.

Response 21: We agree that the introduction of the archetypes is repeated in the discussion, which is not necessary. We have removed the second description of the archetypes that were written as a list.

**Comment 22: Lines 356-359, why did not you use longer period, after Sentinel is launched?**

Response 22: The pre-trained model of DeepAqua was trained for use on images between January 2020 and August 2024. Outside of this range, the model is not generalisable and therefore results in poorly predicted water extents. This is because the original training dataset was likely not large or diverse enough to display a range of noise, climatic conditions, radio frequency interference (RFI) or any other effects that may change the backscatter distribution of the SAR image, which appears to be the case for dates before 2020 and after August 2023. Therefore, we restrict our temporal range to January 2020 – August 2023 to ensure accurate water extent predictions. It would have been ideal to have had a more generalisable model though, implying a larger dataset (2015-present). However, this requires considerable changes to the current open-source version of DeepAqua that fall beyond the scope of this paper.

We now add the following to the end of the paragraph (lines 433-437): "Since the DeepAqua model we used for water extent predictions was trained to predict water extent on SAR scenes dating between January 2020 and August 2023, we were not able to extend our temporal scope outside of this range. Therefore, we suggest developing any future training of the DeepAqua model so that it is more generalisable to longer time periods and less sensitive to changes in Sentinel-1 SAR pre-processing."

Comment 23: For section 4.1, lines 343-370, the logical relationship between the three paragraphs is loose, and the transition between paragraphs is not natural enough. It is recommended to add clearer transition sentences between paragraphs to help readers better understand the connection between the various parts.

Response 23: We agree that the logic between the paragraphs in Section 4.1 feels too choppy, and in general the discussion sections felt a bit 'clunky'. Hence, we have improved the flow of Discussion section 4.1. by firstly highlighting the value of using archetypes based on the hydrological regime for wetland studies, and then secondly addressing potential issues of using archetypes as a type of classification. We then restructure Section 4.2. to only include methodological considerations and then we split up the remaining discussion into Section 4.3. 'Controls and variability in wetland hydrological behaviour' and Section 4.4. 'Hydrological regimes as indicators of ecosystem services'.

**Appendix**

Comment 24: Figure A1-A8 is not referred in the main text part

Response 24: Thank you for noticing this. We have referenced Figure A1 on line 194 and Figures A2-A6 on 154.

Comment 25: Figure A2, what is the meaning of this figure? I did not see anything related in the manuscript.

Response 25: We agree with this comment, therefore we have removed Figure A2 from the appendix.

Comment 26: Figures A3-A8, all sub-figures have a blank box on the top-right.

Response 26: The blank box has now been removed from Figures A3-A8 (now A2-A6).

**General problems**

Comment 27: Some figures are overly large and could be resized for better integration into the text.

Response 27: We have resized all figures so that they are better integrated into the text.

Comment 28: In lines 495-496, the link is invalid (https://github.com/melqkiades/deepwetlands)

Response 28: Thank you for checking the validity of the links. We have now fixed the link by replacing the invalid link with the following: <a href="https://github.com/melqkiades/deep-wetlands">https://github.com/melqkiades/deep-wetlands</a>

**References**

Brown, C. F., Brumby, S. P., Guzder-Williams, B., Birch, T., Hyde, S. B., Mazzariello, J., Czerwinski, W., Pasquarella, V. J., Haertel, R., Ilyushchenko, S., Schwehr, K., Weisse, M., Stolle, F., Hanson, C., Guinan, O., Moore, R., and Tait, A. M.: Dynamic World, Near real-time global 10 m land use land cover mapping, Sci. Data, 9, 251, https://doi.org/10.1038/s41597-022-01307-4, 2022.

McFeeters, S. K.: The use of the Normalized Difference Water Index (NDWI) in the delineation of open water features, Int. J. Remote Sens., 17, 1425–1432, https://doi.org/10.1080/01431169608948714, 1996.

Peña, F. J., Hübinger, C., Payberah, A. H., and Jaramillo, F.: DeepAqua: Semantic segmentation of wetland water surfaces with SAR imagery using deep neural networks without manually annotated data, Int. J. Appl. Earth Obs. Geoinformation, 126, 103624, https://doi.org/10.1016/j.jag.2023.103624, 2024.

**AC2**

We sincerely thank the Editor, Associate Editor and Reviewers for handling and taking the time to read and review our manuscript. We are also grateful for the reviewer's suggestions and we believe the manuscript and the discussion in particular is much stronger as a result. Below are our replies to the comments given:

**Main comments**

Comment 1: The authors may need to more carefully justify their interpretations of ecosystem services and regime stability—i.e., by supporting these interpretations with appropriate references or clearly flagging them as hypotheses.

Response 1: Thank you for this valid and important comment. We agree that our interpretations of ecosystem services are not clearly flagged as hypotheses in the original manuscript; they indeed are not related to the main objective of the study which is to categorise wetlands based on their hydrological regime from water extent observations. We have now clarified in the text that these interpretations are a discussion that requires further investigation falling beyond the scope of this manuscript.

We still believe that linking hydrological regimes to ecosystem service delivery is a helpful and innovative way to interpret our results. To support the discussion of such interpretations, we have complimented the literature already present in the manuscript with site-specific information and additional references (e.g. Okruszko et al., 2011; Doherty et al., 2014) and discuss their findings accordingly in relation to the manuscript.

Okruszko et al., 2011, who assess the impact of future hydrological conditions on the delivery of a range of biotic and abiotic ecosystem services, state that wetland hydrology is a major driving variable for multiple ecosystem services, such as supporting bird populations, wetland vegetation, carbon storage and nutrient removal. Additionally, Doherty et al., 2014 suggest that wetlands with high infiltration capacity and periodically dry soils can slow down flows and remove large volumes of water from the system. We have integrated these references into the discussion accordingly.

We also use the Ramsar site information reports to support our discussion. In particular, we suggest that headwater wetlands such as those classified within the spring-surging archetype, do not typically contribute to flood control. None of these sites list flood control as a prevalent ecosystem service reported by Ramsar in the 'Site Summary'. Building on this, we elaborate on the fact that hydrological regimes such as those of slow-drying wetlands are more analogous to floodplain wetlands, which Bullock and Acreman (2003) describe as providing services of flood attenuation and storage capacity. Furthermore, we mention that over half of the wetlands in

the slow-drying archetype list either water storage or flood attenuation as key ecosystem services based on the Ramsar site summary reports.

Comment 2: In addition, I suggest including an analysis or discussion of interannual variability to assess regime stability, or at least acknowledging the potential influence of climate variability.

Response 2: Thank you for your suggestion. In order to assess regime stability, we calculated the monthly standard deviation from all years from the mean, which we interpret as the inverse of regime stability, i.e., a higher standard deviation implies reduced stability. We have now updated figures A2-A6 to include the degree of regime stability as grey areas of variability around the mean, as well as showing the total standard deviation in small bar plots in the top right of the subplots for each wetland. For instance, the regime stability for the slow-drying wetlands is visualised in the figure below. We also discuss regime stability and hydrological regime shifts in wetlands in Discussion section 4.1.

Figure 1 (Figure A5 in the manuscript). Average monthly water extent (March-October) between 2020-2023 for all wetlands belonging to the slow-drying archetype. Grey area shows the monthly interannual variability given by the range of water extent from all years. The monthly standard deviation is given in the top right bar plots.

We also wanted to acknowledge the influence of climate on the wetland's hydrological regime. Our time series of water extent (~4 years) is indeed too short to investigate the influence of climate on surface water extent, however, we now include precipitation data along with that of water extent. We now plot the daily precipitation in the wetland's watershed (using the Copernicus Climate Change Service E-OBS 0.1-degree daily precipitation dataset using surface

observations) versus water extent for all matching dates (See example below for the spring-surging wetlands archetype). The new version of the manuscript includes observations of daily precipitation plotted against surface water extent for all wetlands in figures A7-A11 in the appendix.

Finally, in addition, we now explicitly acknowledge the influence of climate variability on wetland water extent changes by mentioning other studies which show how hydroclimatic changes and changes in atmospheric water inputs can be responsible for changes in wetland water conditions on lines 490-506. (Jaramillo et al., 2018; Winter, 2000; Xi et al., 2021; Xu et al., 2024; Zhang et al., 2011)

Figure 2 (Figure A7 in the manuscript). Wetland water extent from January 2020 to August 2023 (excluding January, February November and December) for spring-s surging wetlands, shown alongside daily precipitation totals for matching dates. Precipitation is aggregated separately for each wetland's catchment and Ramsar area.

**Comment 3: I recommend the authors incorporate a few more recent and synthetic references, particularly from the past five years:**

Wood, Kevin A., et al. "A global systematic review of the cultural ecosystem services provided by wetlands." Ecosystem Services 70 (2024): 101673.

Mupepi, O., Marambanyika, T., Matsa, M. M., & Dube, T. (2024). A systematic review on remote sensing of wetland environments. Transactions of the Royal Society of South Africa, 79(1), 67–85.

Davidson, Nick C., et al. "Worth of wetlands: revised global monetary values of coastal and inland wetland ecosystem services." Marine and Freshwater Research 70.8 (2019): 1189-1194.

Response 3: We thank the reviewer for suggesting interesting and relevant references that we have now incorporated in the text in lines 37, 452 and 493, respectively.

**Minor comments**

Comment 4: Line 17 the use of "... between 2020-2023." should be "between 2020 and 2023" or "from 2020 to 2023".

Response 4: Thanks, this has been updated to from January 2020 to August 2023.

Comment 5: Line 80, "Doing so would help quantify their ecosystem services (unknown to date), particularly emphasising hydrology-based services such as flood attenuation and low flow supply." "This approach helps quantify their, as yet largely unknown, ecosystem services—particularly those related to hydrology, such as flood attenuation and low-flow support."

Response 5: We have accepted your suggestion for the sentence structure.

Comment 6: Line 182, "...worked together to form to capture..." should be "...worked together to capture..."

Response 6: We have accepted your suggestion for the wording error.

Comment 7: Line 211, "...from which drying occurs after that..." should be "...after which drying occurs."

Response 7: We have accepted your suggestion for the wording error.

Comment 8: Line 349, "...complimented..." should be "complemented"

Response 8: We have accepted your suggestion for the spelling error.

Comment 9: Line 411, "although more data is required" should be "although more data are required".

Response 9: We have accepted your suggestion on the wording error.

Comment 10: Line 444, "Spring-flooded wetlands. Mire and fjäll wetlands found mainly in northern Sweden with a wet period during the Spring that precede a prolonged dry period beginning in June." should be "...Spring-flooded wetlands: mire and fjäll wetlands mainly in

northern Sweden, with a wet period during spring that is preceded by a prolonged dry phase starting in June..."

Response 10: This line has now been removed from the manuscript, as it was a repetition from results section 4.2, so there is no need to accept the suggestion.

**References**

Doherty, J. M., Miller, J. F., Prellwitz, S. G., Thompson, A. M., Loheide, S. P., and Zedler, J. B.: Hydrologic Regimes Revealed Bundles and Tradeoffs Among Six Wetland Services, Ecosystems, 17, 1026–1039, https://doi.org/10.1007/s10021-014-9775-3, 2014.

Jaramillo, F., Licero, L., Åhlen, I., Manzoni, S., Rodríguez-Rodríguez, J. A., Guittard, A., Hylin, A., Bolaños, J., Jawitz, J., Wdowinski, S., Martínez, O., and Espinosa, L. F.: Effects of Hydroclimatic Change and Rehabilitation Activities on Salinity and Mangroves in the Ciénaga Grande de Santa Marta, Colombia, Wetlands, 38, 755–767, https://doi.org/10.1007/s13157-018-1024-7, 2018.

Jing, L., Zeng, Q., He, K., Liu, P., Fan, R., Lu, W., Lei, G., Lu, C., and Wen, L.: Vegetation Dynamic in a Large Floodplain Wetland: The Effects of Hydroclimatic Regime, Remote Sens., 15, 2614, https://doi.org/10.3390/rs15102614, 2023.

Okruszko, T., Duel, H., Acreman, M., Grygoruk, M., Flörke, M., and Schneider, C.: Broad-scale ecosystem services of European wetlands—overview of the current situation and future perspectives under different climate and water management scenarios, Hydrol. Sci. J., 56, 1501–1517, https://doi.org/10.1080/02626667.2011.631188, 2011.

Wen, L., Macdonald, R., Morrison, T., Hameed, T., Saintilan, N., and Ling, J.: From hydrodynamic to hydrological modelling: Investigating long-term hydrological regimes of key wetlands in the Macquarie Marshes, a semi-arid lowland floodplain in Australia, J. Hydrol., 500, 45–61, https://doi.org/10.1016/j.jhydrol.2013.07.015, 2013.

Winter, T. C.: The Vulnerability of Wetlands to Climate Change: A Hydrologic Landscape Perspective1, JAWRA J. Am. Water Resour. Assoc., 36, 305–311, https://doi.org/10.1111/j.1752-1688.2000.tb04269.x, 2000.

Xi, Y., Peng, S., Ciais, P., and Chen, Y.: Future impacts of climate change on inland Ramsar wetlands, Nat. Clim. Change, 11, 45–51, https://doi.org/10.1038/s41558-020-00942-2, 2021.

Xu, D., Bisht, G., Tan, Z., Sinha, E., Di Vittorio, A. V., Zhou, T., Ivanov, V. Y., and Leung, L. R.: Climate change will reduce North American inland wetland areas and disrupt their seasonal regimes, Nat. Commun., 15, 2438, https://doi.org/10.1038/s41467-024-45286-z, 2024.

Zhang, H., Huang, G. H., Wang, D., and Zhang, X.: Uncertainty assessment of climate change impacts on the hydrology of small prairie wetlands, J. Hydrol., 396, 94–103, https://doi.org/10.1016/j.jhydrol.2010.10.037, 2011.

---

## Author Response (AR2)

**Editor decision: Publish subject to minor revisions (review by editor)**

Dear authors.

The reviewers have made some technical suggestions to your revision.

I invite you consider them and submit a new revision for my consideration.

Best wishes

Bob Su

Thank you for the opportunity to revise our manuscript. We appreciate the constructive feedback provided by the reviewers, which has helped us to improve the clarity, transparency, and robustness of our study. Below, we provide detailed responses to each comment and outline the corresponding amendments made to the manuscript.

**Report #1**

Comment 1. Comment 7: I could not find citation in line 119 in version 2, only saw it in tracked change version.

Response 1. We have now cited the figure on line 119.

Comment 2. The tracked change version and the version 2 is not the same. For example, in lines 18-19 in version 2, and lines 19-20 in tracked change version. The first one is "five archetypes", but the second file has six archetypes. And I saw the analysis and figures in both versions are five archetypes. Please could you confirm this? Because in version 1, there was six archetypes. I could not find any comments or explanation why you changed to five archetypes. If it's correct, please check the final version before publish, make sure everything is consistent.

Response 2. The reviewer is correct that the revised manuscript presents five archetypes, whereas the earlier version included six. This change arose from re-running the cluster analysis during the first review round. In that process, we identified that DeepAqua underestimated wetland water extent for some wetlands between August and October 2023, which led to around 12 wetlands being misclassified as 'autumn drying wetlands'. After removing these affected dates and repeating the cluster analysis, this archetype was no longer present, and the optimal number of clusters was reduced to five. We have carefully ensured that all figures, tables, and text are consistent with this updated result.

**Report #2**

Comment 3. This is an innovative and timely paper that proposes a globally applicable hydrological classification of wetlands. The concept of archetypes offers a useful tool for understanding and comparing wetland regimes. However, several areas, particularly methodology transparency, contextual comparisons, and limitation acknowledgment, need to be strengthened. I recommend major revision before publication.

Response 3. Thank you. Please find below the improvements that we have applied concerning each of the "areas" mentioned by the reviewer in Comment 3:

**Methodology transparency** - To improve clarity in our site filtering, we now define 'low SAR data availability' and outline its causes (Lines 97-99). For the surface water detection, we added the DeepAqua model version and noted that no fine-tuning was performed, ensuring that the same openly available model can reproduce our results (Lines 146-147). We also clarified the source and preprocessing of the Sentinel-1 SAR imagery on Google Earth Engine. (Lines 154-156), ensuring results can be directly replicated.

For validation, we describe the site selection (Lines 168-170) and confirm that delineation was conducted independently of DeepAqua predictions by an experienced interpreter (Lines 170-171). In the cluster analysis (Subsection 2.4), we also specify setting a random seed of 42 for reproducibility (Lines 183-184).

Finally, the supplementary information now includes an Excel file listing all SAR scene IDs per wetland, and we emphasise that all data are openly available on Zenodo (Line 162-163). Likewise, all processing code is openly available on GitHub, as noted in the manuscript assets.

Contextual comparisons – To strengthen contextual comparisons of hydrological regime and their links to ecosystem services, we have added and re-emphasised key citations. We now reference Lane et al. (2018) and Matti et al. (2017) to support the occurrence of slow-drying wetlands in temperature landscapes and Swedish catchments (Lines 480-484). To highlight the role of hydroclimatic variability, we include Park et al. (2022) and Prigent et al. (2001), which distinguish snow-affected northern Sweden from rain-fed central and southern regions (Lines 489-493).

We expand on the flood control role of spring-surging wetlands with a case study of wetland-rich headwaters in central Europe (Votrubova et al., 2017; Lines 523-525) and add Opperman et al. (2010) to support flood attenuation by floodplains (Line 530). Finally, we re-emphasise Åhlén et al. (2022), showing that downstream wetlands in central Sweden remain relatively dry and retain their flood-buffering capacity compared to headwater wetlands (Lines 532-535).

**Limitation acknowledgment** - To more clearly acknowledge the limitations of our study, we added a dedicated subsection (4.2: Methodological considerations). In addition to noting possible improvements, we now explicitly state that our observation period does not capture the full range of long-term hydrological variability (Lines 438-439).

The paragraph on Lines 442-451 has been revised to focus on sensor-related limitations, emphasising the exclusion of smaller wetlands and suggesting future use of longer-wavelength SAR (e.g., NISAR). A new

paragraph (Lines 453-458) discusses the constraints of using DeepAqua, including why predictions could not extend beyond our study period and how the model could be improved. Finally, we acknowledge (Lines 467-471) that incorporating additional variables could strengthen the explanatory power of the archetypes and address winter observational gaps.

**References**

Åhlén, I., Thorslund, J., Hambäck, P., Destouni, G., and Jarsjö, J.: Wetland position in the landscape: Impact on water storage and flood buffering, Ecohydrology, 15, e2458, https://doi.org/10.1002/eco.2458, 2022.

Lane, B. A., Sandoval-Solis, S., Stein, E. D., Yarnell, S. M., Pasternack, G. B., and Dahlke, H. E.: Beyond Metrics? The Role of Hydrologic Baseline Archetypes in Environmental Water Management, Environ. Manage., 62, 678–693, https://doi.org/10.1007/s00267-018-1077-7, 2018.

Matti, B., Dahlke, H. E., Dieppois, B., Lawler, D. M., and Lyon, S. W.: Flood seasonality across Scandinavia—Evidence of a shifting hydrograph?, Hydrol. Process., 31, 4354–4370, https://doi.org/10.1002/hyp.11365, 2017.

Opperman, J. J., Luster, R., McKenney, B. A., Roberts, M., and Meadows, A. W.: Ecologically Functional Floodplains: Connectivity, Flow Regime, and Scale, JAWRA J. Am. Water Resour. Assoc., 46, 211–226, https://doi.org/10.1111/j.1752-1688.2010.00426.x, 2010.

Park, J., Kumar, M., Lane, C. R., and Basu, N. B.: Seasonality of inundation in geographically isolated wetlands across the United States, Environ. Res. Lett., 17, 054005, https://doi.org/10.1088/1748-9326/ac6149, 2022.

Prigent, C., Matthews, E., Aires, F., and Rossow, W. B.: Remote sensing of global wetland dynamics with multiple satellite data sets, Geophys. Res. Lett., 28, 4631–4634, 2001.

Votrubova, J., Dohnal, M., Vogel, T., Sanda, M., and Tesar, M.: Episodic runoff generation at Central European headwater catchments studied using water isotope concentration signals, J. Hydrol. Hydromech., 65, 114–122, https://doi.org/10.1515/johh-2017-0002, 2017.